# Evaluating the Efficacy of Machine Performance Checks as an Alternative to Winston–Lutz Quality Assurance Testing in the TrueBeam Linear Accelerator with HyperArc

**DOI:** 10.3390/diagnostics14040410

**Published:** 2024-02-13

**Authors:** Eun Kyu Kim, Sung Yeop Kim, Jae Won Park, Jaehyeon Park, Ji Woon Yea, Yoon Young Jo, Se An Oh

**Affiliations:** 1Department of Physics, Yeungnam University, Gyeongsan 38541, Republic of Korea; ralisakim@gmail.com (E.K.K.); tgs01178@gmail.com (S.Y.K.); 2Department of Radiation Oncology, Yeungnam University Medical Center, Daegu 42415, Republic of Korea; kapicap@ynu.ac.kr (J.W.P.); drjhyeon@ynu.ac.kr (J.P.); yjw1160@ynu.ac.kr (J.W.Y.); cantabile801@naver.com (Y.Y.J.); 3Department of Radiation Oncology, Yeungnam University College of Medicine, Daegu 42415, Republic of Korea

**Keywords:** isocenter, HyperArc technique, brain metastasis, machine performance check, Winston–Lutz test, TrueBeam linear accelerator

## Abstract

HyperArc is a preferred technique for treating brain metastases, employing a single isocenter for multiple lesions. Geometrical isocentricity in the TrueBeam linear accelerator with HyperArc is crucial. We evaluated machine performance checks (MPCs) as an alternative to the Winston–Lutz (WL) test to verify the treatment isocenter. Between January and July 2023, we assessed 53 data points using MPC and Winston–Lutz tests. The isocenter size obtained from the MPC and its sum, including the rotation-induced couch shift, were compared with the maximum total delta value from the Winston–Lutz test. The maximum total delta was 0.68 ± 0.10 mm, while the isocenter size was 0.28 ± 0.02 mm. The sum of the isocenter size and rotation-induced couch shift measured by MPC was 0.61 ± 0.03 mm. During the Winston–Lutz test (without couch rotation), the maximum total delta value was 0.56 ± 0.13 mm. A *t*-test analysis revealed a significant difference in the isocenter size averages between the Winston–Lutz and MPC outcomes, whereas the Pearson’s correlation coefficient yielded no correlation. Our study highlights the necessity for separate MPC and Winston–Lutz tests for isocenter verification. Therefore, the Winston–Lutz test should precede stereotactic radiosurgery for isocenter verification.

## 1. Introduction

Brain metastases (BMs), which are 10 times more prevalent than primary brain tumors, constitute the most common intracranial neoplasms [1,2,3,4], affecting 20–40% of cancer patients and having frequent neurological complications [5,6,7]. The global incidence rate of BMs continues to rise, posing a significant challenge to cancer care [8]. Treatment options include surgical resection, whole-brain radiotherapy (WBRT), stereotactic radiosurgery (SRS), chemotherapy, or a combination of these [2,9,10]. Therefore, treatment selection requires careful consideration of tumor size, general systematic performance, neurological function, and tumor type [8].

SRS, which administers a concentrated dose in a single session to a precisely targeted small intracranial area, has emerged as the leading neurosurgical approach [5,8,11,12,13]. Kondziolka et al. reported that combining WBRT and SRS significantly enhances brain disorder control and survival rates compared with WBRT [14]. Tsao et al., from the American Society for Radiation Oncology Guideline Task Group, published a guideline for brain metastasis management: radiotherapeutic and surgical management for newly diagnosed brain metastasis(es), with SRS as the primary choice for selected patients with an expected survival period of three to twelve months [15]. Therefore, SRS is considered the best treatment for multiple BMs because quality of life and long-term survival are the primary treatment objectives [1,8].

However, conventional SRS requires extensive planning efforts and patient setup for individual isocenters, with separate planning for each lesion. Recently, the volumetric modulated arc therapy (VMAT) technique has emerged as a solution for treating multiple brain lesions using a single isocenter with single or multiple arcs [2,9,16,17]. In 2017, the HyperArc VMAT (HA-VMAT, Varian Medical Systems, Palo Alto, CA, USA) plan was clinically implemented as a new solution to supplement SRS dose delivery demands [5,18]. This innovative module facilitates automated noncoplanar intracranial SRS treatments with collision-free single isocenter delivery and steep dose gradients [18,19,20].

Treatment precision and geometric accuracy are vital when managing patients with multiple BMs [16,19,21]. SRS demands a high degree of positioning accuracy, a steeper dose gradient, and meticulous dose verification [11]. The American Association of Physicists in Medicine (AAPM) Task Group (TG)-142 and TG-179 provide guidelines for quality assurance (QA) tests of linear accelerators and computer tomography (CT)-based image-guided radiation therapy (IGRT) systems, including kilovolt and megavolt cone-beam CT, fan-beam MVCT, and CT-on-rails [22]. To avoid dose errors and deviations, the coincidence of the radiation and mechanical isocenter for SRS/SBRT machines should be maintained within a tolerance of ±1 mm from the baseline [23].

Daily checks with a tolerance of ±2 mm are recommended for laser/image/treatment isocenter coincidence or phantom localization and repositioning through couch shift. Nonetheless, monthly checks with a tolerance of ±1 mm ensure the accuracy of MV/kV/laser alignments and couch shifts to measure the geometric accuracy.

Machine performance checks (MPCs), introduced in 2013 by Varian (Varian Medical Systems, Palo Alto, CA, USA), comprise a TrueBeam 2.0 application for the automated verification of geometric and beam performances via kV-MV imaging systems. Numerous institutions adopt the MPC for daily testing, and its performance has been extensively evaluated by various studies [4,24,25,26]. However, few studies have compared MPCs with standard QA tests to determine whether MPCs can replace them. In the studies by Clivio et al. [24] and Barnes et al. [25], the results of MPCs were compared to those of conventional QA tests.

Recently, Clivio et al. compared the geometry and beam performances of TrueBeam LINAC using MPCs, with alternative, independent checks performed routinely [24]. They evaluated > 10 repetitions on consecutive days and employed the Winston–Lutz test for isocenter radius comparison in the geometric check. The study concluded that the MPC and Winston–Lutz test results were consistent in terms of the isocenter size and position.

Barnes et al. evaluated the performance of MPC isocenter and couch tests compared with in-house-developed Winston–Lutz and routine QA tests [25]. They demonstrated that MPC kV imager offset could achieve daily isocenter alignment, and less frequent Winston–Lutz or isocenter verification could be used for assurance. However, both studies employed an in-house Winston–Lutz test analysis and the acquired data. Therefore, our study aims to provide a long-term stability assessment using DoseLab analysis of the Winston–Lutz test compared with the MPC results. Additionally, motivated by the hypothesis that the MPC can replace the Winston–Lutz test to ensure isocenter coincidence and position, this study aims to verify this substitution by geometrically checking the treatment isocenter size on the TrueBeam LINAC.

## 2. Materials and Methods

This study was conducted between January 2023 and July 2023, during which 53 data points were collected. All measurements were obtained using a 6 MV beam on a TrueBeam LINAC (Varian Medical Systems, Palo Alto, CA, USA) with ARIA OIS (Version 15.6, Varian Medical Systems) at the Yeungnam University Medical Center. MPCs were conducted each morning before patient treatment, while the Winston–Lutz test was performed in sequence. The MPC and Winston–Lutz tests both required approximately 15 min.

### 2.1. Winston–Lutz Test

In radiation therapy, the Winston–Lutz test is commonly used to verify the isocenter position and size of a linear accelerator [27]. This procedure is crucial before SRS treatment, because it ensures the geometric accuracy of the radiation beam, confirming its accurate delivery to the target center of the lesion and detecting any displacements in the isocenter position [28]. Figure 1 shows a schematic of the Winston–Lutz geometry. The isocenter, as defined by the lasers in the treatment room, was identified by placing a ball bearing at its position. Imaging was performed by irradiating the beam at different angles of the gantry, collimator, and couch using an electronic portal imaging device (EPID). Our study collected eight images from the irradiating beams at different angle combinations (see Table 1). A WL^3^ (6.35 cm × 6.35 cm × 6.35 cm cube-shaped) phantom designed for DoseLab Winston–Lutz analysis (Varian Medical Systems), which contained a 5 mm tungsten sphere at its center, was used for the isocenter arrangement (Figure 2a). The EPID image analysis was conducted using the DoseLab Pro software (Version 7.0MR1, Varian Medical Systems). The offset between the center of the ball and the center of the radiation field was calculated for each image associated with isocenter movements. The delta value indicates this difference in length (mm) between the centers of the ball and radiation field. The results show the maximum delta, maximum total delta, mean total delta, and three-dimensional target positions on the *x*-, *y*-, and *z*-axes. The maximum delta represents the maximum absolute horizontal or vertical two-dimensional delta between the center of the radiation field and the center of the ball. The maximum total delta corresponds to the maximum value among the total distances of all images. The total distance was calculated by adding the horizontal and vertical offsets of the two centroids in quadrature [29]. This definition aligns with the definition of the isocenter size measured by the MPC. Therefore, our study considered the maximum total delta as a comparative value.

### 2.2. Machine Performance Check

An MPC (Version 2.22, Varian Medical Systems, Palo Alto, CA, USA) was employed to verify geometric and beam properties by automatically processing numerous MV and kV images obtained for divergent gantry, collimator, and couch settings. The IsoCal phantom was placed on a couch with its phantom holder for MPC performance (Figure 2b). The MPC application supports both beam constancy and geometric checks, enabling the assessment of the size of the treatment isocenter and its coincidence with the imaging isocenter, the accuracy of the kV and MV imaging system position, collimator accuracy, gantry rotation angle, couch position, jaw accuracy, and MLC leaf position. Among the various mechanical components, the isocenter, defined as the ideal intersection point of the central axis of the beam during the full gantry rotation, plays a crucial role in ensuring accurate target localization and treatment planning. The MPC geometric check determined the isocenter size as the maximum distance of the central axis of the beam from the idealized isocenter. The central beam axis was defined by the center of the MLC rotation, and the isocenter was determined through the rotation of eight gantry angles (0°, 45°, 90°, 135°, 180°, 225°, 270°, and 315°) [30].

### 2.3. Comparisons of the Winston–Lutz Test and the MPC

#### 2.3.1. Maximum Total Delta Using Winston–Lutz and Isocenter Size Using MPC

The maximum total delta obtained from the Winston–Lutz test and the isocenter size measured by the MPC share a congruent definition, referring to the maximum distance between the radiation isocenter and the center of the axis. Given that both methods aim to measure the treatment isocenter size and verify isocenter localization for QA, we endeavored to establish a correlation by comparing the results of each method.

#### 2.3.2. Maximum Total Delta Using Winston–Lutz Compared with the Sum of Isocenter Size and Rotation-Induced Couch Shift Using MPC

The rotation-induced couch shift parameter acquired from the MPC is obtained from a geometric check of the couch positioning accuracy and indicates the discrepancy between the center of rotation and the treatment isocenter. The center of the couch rotation was identified as a distinct location from the treatment isocenter, as determined by the couch rotation on all available rotational axes: lateral, longitudinal, vertical, rotational, pitch, and roll. Consequently, when the couch was rotated, the isocenter was displaced from the initial position and quantified using the phantom set at a couch angle of 0°. The sum of the isocenter size and rotation-induced couch shift parameter aims to consider the isocenter displacement caused by couch rotation and becomes a relevant factor in the correlation analysis.

#### 2.3.3. Maximum Total Delta without Couch Rotation Using Winston–Lutz and Isocenter Size Using MPC

Another set of statistical analyses was performed using data collected from cases without couch rotation during the Winston–Lutz test, focusing solely on gantry and collimator rotation. Six EPID images were analyzed for each Winston–Lutz process, all acquired with a couch angle of 0°. As the central beam axis experienced variations due to changes in the gantry angle, the results were compared to identify differences in measurements, considering cases with couch rotation. Furthermore, an attempt was made to establish a correlation when the target remained unaffected by couch movement.

#### 2.3.4. Statistics

To evaluate the mean difference between two independent continuous datasets, a one-sample *t*-test was utilized, with a significance threshold set at *p* < 0.05. This statistical test was applied in our study to assess the hypothesis that average measurements obtained from the Winston–Lutz and MPC tests were equivalent. The calculated *t*- and *p*-values specifically represent the differences in the average maximum total delta and the size of the isocenter. This analysis also included scenarios involving rotation-induced couch shifts. Additionally, a Pearson correlation analysis was conducted to measure the linear relationship. The correlation coefficient ranged from −1 to 1, where values close to 1 or −1 indicated a strong positive or negative correlation, respectively. As the correlation coefficient approached zero, the linear relationship weakened.

## 3. Results

### 3.1. Isocenter Size Using the MPC

This study analyzed 53 data points and compared the maximum total delta obtained from the Winston–Lutz test with the DoseLab analysis and the isocenter size measured using the MPC. The maximum total delta ranged from 0.46 to 0.88 mm, with a mean of 0.68 ± 0.10 mm. However, the isocenter size measured by MPC exhibited less variation, ranging from 0.22 to 0.33 mm, with a mean of 0.28 ± 0.02 mm (Table 2). Figure 3a illustrates the overall range and mean differences between the two results. A single-sample *t*-test was performed to compare the means statistically. The results showed that the two methods did not agree statistically (t(52) = 28.130, *p* < 0.001). Furthermore, a Pearson correlation analysis was performed, which yielded a correlation coefficient (r) of −0.005 (*p* = 0.969), thereby indicating a negligible correlation (Figure 4a).

### 3.2. Sum of the Isocenter Size and the Rotation-Induced Couch Shift Using the MPC

Figure 3b shows the difference in range between the isocenter size with rotation-induced couch shift using the MPC and the maximum total delta using the Winston–Lutz test. The mean (± standard deviation) of the isocenter size with rotation-induced couch shift using the MPC over the total period was 0.61 ± 0.03 mm and ranged from 0.55 to 0.67 mm. However, the mean difference from the maximum total delta using the Winston–Lutz was 0.07 mm, indicating a smaller value disparity. This analysis shows the least difference based on the two methods among all the conducted analyses. The *t*-test analysis also demonstrated statistical disagreement compared to (t(52) = 4.776, *p* < 0.001), and the Pearson correlation coefficient (r) was 0.043 (*p* = 0.759), which indicates a negligible correlation between the two results (Figure 4b).

### 3.3. Comparison without Couch Rotation in Winston–Lutz Test

In the Winston–Lutz test conducted without couch rotation, the maximum total delta exhibited an average of 0.56 ± 0.13 mm, with a range extending from 0.34 to 0.88 mm. This finding differs from the results obtained with couch rotation, where both the minimum range value and the mean decreased by 0.12 mm. Conversely, the isocenter size measured in the MPC test yielded a mean of 0.28 ± 0.02 mm, spanning a range of 0.22–0.33 mm. Comparing the maximum total delta and isocenter size, an average discrepancy of 0.28 mm was observed between the two methods. The *t*-test and correlation analysis results are consistent with the case involving couch rotation, that is, (t(52) = 15.336, *p* < 0.001), thereby indicating statistical significance. Lastly, the correlation coefficient (r) was −0.069, which indicates almost no correlation between the two sets of results.

## 4. Discussion

This study investigated the feasibility of using the Winston–Lutz test as an alternative to the MPC method for measuring isocenter sizes in QA tests. We compared the average isocenter size measured by the MPC with the mean of the maximum total delta obtained from the Winston–Lutz test using statistical analysis. The study also considered scenarios involving the inclusion of couch shift owing to rotations in isocenter size measurements, as well as instances where the Winston–Lutz test was conducted without couch rotation. The one-sample *t*-test showed statistical significance, thus indicating that there was a difference between the means of the two methods (see Table 3). However, the correlation analysis consistently demonstrated an insignificant correlation. The mean difference reached a minimum value of 0.05 mm when the sum of the isocenter size and the rotation-induced couch shift parameter were compared with the maximum total delta. Nonetheless, some data points showed significant differences that exceeded 0.2 mm. The primary comparison between the maximum total delta using Winston–Lutz and the isocenter size using the MPC displayed broad, dispersed data values that exhibited excessive differences.

Although numerous studies have evaluated MPC performance or assessed isocenter localization, limited reports have directly compared the isocenter sizes of the MPC and the Winston–Lutz test using the DoseLab software over an extended period.

A similar comparison was performed by Barnes et al. [25], who examined the MPC isocenter and couch tests for routine QA evaluations. They used the Winston–Lutz test to determine the isocenter position and size using an in-house-developed MATLAB script for EPID image analysis. The *t*-test demonstrated a lack of statistical agreement between the two methods, similar to the findings of our study. However, the 95% confidence interval [0.34, 0.39] of the Winston–Lutz data mean included the MPC data mean during the measurements of the isocenter size. The Winston–Lutz data were consistent with the MPC data, with a similar range and mean values of 0.37 ± 0.06 (WL) and 0.34 ± 0.02 mm (MPC). Additionally, data from both methods were within the agreement of ±0.2 mm, which indicated clinical accordance.

Clivio et al. [24] evaluated the MPC on a TrueBeam LINAC and performed an isocenter check using the Winston–Lutz test with IsoLock procedure and a MarkerBlock phantom from Varian. Their study, which involved more than 11 acquisitions with a 6 MV beam, reported an isocenter size mean of 0.34 ± 0.01 mm and 0.264 mm using the MPC and Winston–Lutz tests, respectively, resulting in a mean difference of 0.076 mm. When measuring the rotation-induced couch shift, the Winston–Lutz and MPC tests yielded a shift of 0.803 mm and 0.37 ± 0.02 mm, respectively. Each parameter was compared individually, whereas our study considered combining the couch shift with the isocenter size. The mean rotation-induced couch shift from the MPC was 0.33 ± 0.02 mm; therefore, further investigation of the couch shift by the Winston–Lutz test is required for an accurate comparison.

The *t*-test results of the studies by Barnes et al. and Clivio et al. were consistent with those of our study, indicating statistical significance. However, all studies reported lower isocenter sizes and smaller standard deviations than our result when performing the Winston–Lutz test. The isocenter size data collected using MPC appeared consistent, with a deviation of 0.02 mm. However, the standard deviation of the maximum total delta measured by Winston–Lutz was 0.10 mm, indicating fluctuations in the measured values (see Table 4).

This study has several limitations that can impact the accuracy of the results. One major limitation relates to the errors that occurred when the phantom was positioned through the laser in the treatment room. In some cases, the laser was incorrectly positioned, leading to inaccuracies in the data collected during the study period. Consequently, several calculated total delta measurements reached the warning and fail tolerance values, thereby affecting the Winston–Lutz test results. Therefore, the phantom should be positioned after ensuring the appropriate vertical couch position, which is determined by rotating the gantry by 90 or 270°.

Another limitation is the relatively short data acquisition period. Adjustments to the couch stand and turntable were made in January 2023 owing to values repeatedly exceeding the threshold of ±0.75 mm. Hence, the rotation-induced couch shift showed a significant reduction. Therefore, our study analyzed the data assembled only after alignment completion, reducing the total data.

For the Winston–Lutz test, our institution used the couch position angles from the plan’s gantry, collimator, and couch angle specified in the auto QA module provided by Mobius. Couch angles were set to 0° with 90° and 270°, which are the central point and the two extremes of motion. Couch angles 90°, 45°, 0°, 315°, and 270° had to be considered to obtain an optimal value, which is another limitation.

Eagle et al. [31] investigated the outcomes of off-axis Winston–Lutz tests, utilizing DoseLab Pro (Version 6.7) and an EPID on a TrueBeam system. In single-isocenter multi-target SRS, a significant challenge arises from the potential inaccuracy in targeting off-axis targets, which can be attributed to the decrease in accuracy with increasing off-axis distance. Consequently, verifying the precision of off-axis isocenter positioning becomes critical for safely treating multiple off-axis targets. Eagle et al. [31] introduced a method for conducting off-axis Winston–Lutz tests to assess SRS delivery accuracy at various off-axis distances. However, the MPC test cannot provide data regarding the off-axis isocenter. Therefore, while our study effectively verified the machine isocenter, it had limitations in validating the off-axis isocenter.

## 5. Conclusions

This study analyzed the correlation between the isocenter size measured by MPC and the maximum total delta measured by the Winston–Lutz test using DoseLab. The statistical *t*-test and Pearson correlation analysis revealed no correlation between the two methods. Overall, the MPC parameters displayed lower mean values and deviations compared with the DoseLab results. The Winston–Lutz and MPC tests cannot be used interchangeably. Thus, both methods need to be performed separately. While the MPC is a reliable QA method with clinically meaningful results, the Winston–Lutz test remains crucial and should be regularly performed before SRS to verify the isocenter accuracy.

In the future, different types of Winston–Lutz methods or different analysis software can be used to identify new correlations with the MPC results. In future studies, we will attempt to gather more data and use new methods to discover correlations and predict the outcome values of the Winston–Lutz test.

## Figures and Tables

**Figure 1 diagnostics-14-00410-f001:**
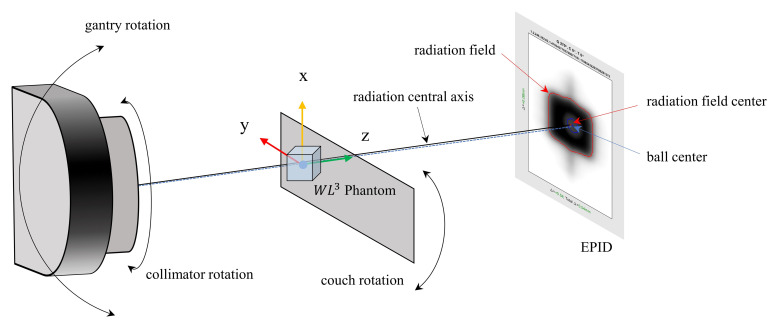
Schematic of Winston–Lutz test geometry using the WL3 phantom. The ball bearing of the phantom is placed at the position of the isocenter by adjusting the couch. The origin of the *x*–*y*–*z* axis was set to the center of the ball bearing. The field edge and center of radiation and ball are displayed in each electronic portal imaging device (EPID) image.

**Figure 2 diagnostics-14-00410-f002:**
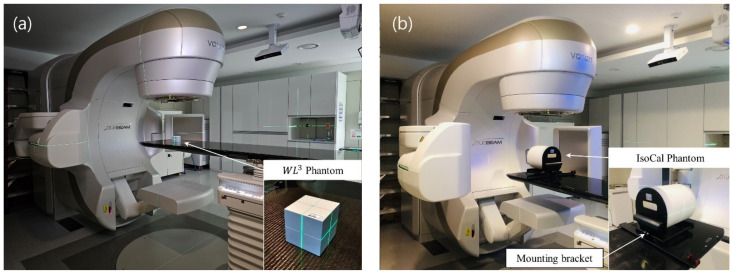
Images of the quality assurance (QA) setup in the treatment room with the TrueBeam linear accelerator. (**a**) Winston–Lutz setup with WL3 phantom and (**b**) machine performance check setup with IsoCal phantom on the mounting bracket.

**Figure 3 diagnostics-14-00410-f003:**
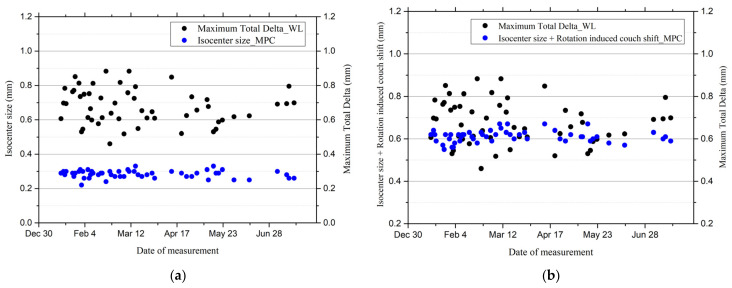
Data distribution of Winston–Lutz and machine performance check (MPC) test results. (**a**) Comparison between the maximum total delta using Winston–Lutz and the isocenter size using the MPC. (**b**) Comparison between the maximum total delta using Winston–Lutz and the isocenter size with rotation-induced couch shift using the MPC.

**Figure 4 diagnostics-14-00410-f004:**
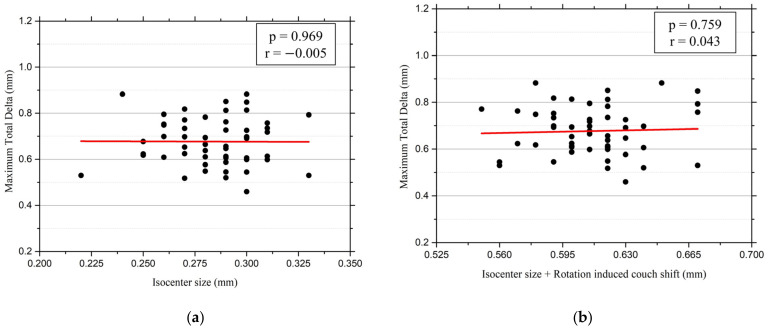
Linearly fitted scatter plots of Pearson correlation analysis between Winston–Lutz and MPC results. (**a**) Correlation between the maximum total delta using Winston–Lutz and isocenter size using the MPC. (**b**) Correlation between maximum total delta using Winston–Lutz and isocenter size with rotation-induced couch shift using the MPC.

**Table 1 diagnostics-14-00410-t001:** LINAC machine parameters for the Winston–Lutz test. Electronic portal imaging device (EPID) images were obtained using eight different gantry, collimator, and couch rotation angle sets when performing the Winston–Lutz test.

No.	Gantry (°)	Collimator (°)	Couch (°)
1	0	0	0
2	0	90	0
3	0	270	0
4	90	0	0
5	180	0	0
6	270	0	0
7	0	0	90
8	0	0	270

**Table 2 diagnostics-14-00410-t002:** Comparative analysis of maximum total delta in Winston–Lutz test and isocenter size in the machine performance check (MPC) test with and without couch rotation components.

Winston–Lutz	Maximum Total Delta (Mean ± SD) (mm)	
	Without couch rotation	With couch rotation
	0.56 ± 0.13	0.68 ± 0.10
**MPC**	**Isocenter Size (Mean ± SD) (mm)**	
	Without rotation-induced couch shift	With rotation-induced couch shift
	0.28 ± 0.02	0.61 ± 0.03

**Table 3 diagnostics-14-00410-t003:** Results of one-sample *t*-tests comparing the means of maximum total delta and isocenter size from Winston–Lutz and MPC tests.

Isocenter Measurement		One-Sample *t*-Test	
MPC	Winston–Lutz Test	Degree of Freedom	*p*-Value
Isocenter size without rotation-induced couch shift	Maximum total delta with couch rotation	52	*p* < 0.001
	Maximum total delta without couch rotation	52	*p* < 0.001
Isocenter size with rotation-induced couch shift	Maximum total delta with couch rotation	52	*p* < 0.001
	Maximum total delta without couch rotation	52	*p* < 0.001

**Table 4 diagnostics-14-00410-t004:** Comparison of results from Barnes et al., Clivio et al., and our study.

	Study Period	MPC Isocenter Size (mm)	Winston–Lutz Isocenter Size (mm)	Difference (WL-MPC)	*t*-Test
Barnes [25]	4 months	0.34 ± 0.02	0.37 ± 0.06	Within ± 0.11 mm	Not in statistical agreement
Clivio [24]	10 repetitions over 3 weeks	0.34 ± 0.01	0.264	0.076 mm	
Our study	4 months	0.28 ± 0.02	0.59 ± 0.14 (maximum total delta)	0.04 mm–0.64 mm	

## Data Availability

The data are available from the authors upon reasonable request. Please contact sean.oh5235@gmail.com with any questions or requests.

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
