# Peer review of "Evaluating the Efficacy of Machine Performance Checks as an Alternative to Winston–Lutz Quality Assurance Testing in the TrueBeam Linear Accelerator with HyperArc"

_diagnostics, 2024, doi:10.3390/diagnostics14040410_

Round 1

Reviewer 1 Report

Comments and Suggestions for Authors

Dear authors,

This is a well written manuscript. Below are my comments that in my point of view may improve the submitted manuscript

Comment 1

In my point of view the novelty of the manuscript in its current form is not stressed enough and it seems more suitable as a technical note 

Comment2

lines 73-74. The authors should reference the related studies. If they mean Clivio et al and Barnes et al mentioned in the following lines they should clarify it in the manuscript, perhaps by using one paragraph for the lines 69-90

Comment3

lines 111-113. Please define delta when it first appears in the manuscript. I assume you mean difference

Comment4

It might be of practical importance to indicate (perhaps in table 3) the time required for each type MPC and Winston-Lutz test

Comment5

lines 254-284. As a tool to better clarifying the novelty of the submitted work, insert a comparable table with yours as well as all the related studies mentioned, indexing what each study considers. Then highlight the difference of your work

with kind regards

Reviewer 2 Report

Comments and Suggestions for Authors

The insight I have is with regard to table 1. The authors mention that they test at couch zero degrees, 90 degrees and 270 degrees, which are the central point and the two extremes of motion. Couch zero is with the couch pointing straight to the gantry. 90 and 270 are with the couch turned 90 degrees to the left or right.

There does appear to have been a determination of appropriate parameters to test the couch motion. Did the authors consider 90, 45, 0, 315 and 270 as an alternative to 90, 0, and 270 degrees couch rotation? If not, this should be mentioned in the discussion as a limitation.

It would also be helpful if the authors would give the reason that they used 90, 0 and 270

Number of data points: 53 is not a lot for a routine test that you would do daily. That's 2 measurements a week over the six month period. So they did not have a stereo patient every day. They do explain that there was an adjustment and that reduced the available period.

Comments on the Quality of English Language

No comments

Round 2

Reviewer 1 Report

Comments and Suggestions for Authors

Dear authors,

thank you for your answers, I am OK

with kind regards